# Novel Design Techniques for the Fermat Spiral in Antenna Arrays, for Maximum SLL Reduction

**DOI:** 10.3390/mi13112000

**Published:** 2022-11-17

**Authors:** Kleiverg Encino, Marco A. Panduro, Alberto Reyna, David H. Covarrubias

**Affiliations:** 1CICESE Research Center, Electronics and Telecommunications Department, Carretera Ensenada-Tijuana No. 3918, Zona Playitas, Ensenada 22860, Baja California, Mexico; 2Electronics Department, Universidad Autónoma de Tamaulipas, UAMRR-R, Carretera Reynosa-San Fernando, Reynosa 88779, Tamaulipas, Mexico

**Keywords:** Fermat spiral, antenna array, side lobe level, amplitude excitation

## Abstract

This paper presents novel design techniques for the Fermat spiral, considering a maximum side lobe level (SLL) reduction. The array system based on a Fermat spiral configuration considers techniques based on uniform and non-uniform amplitude excitation. The cases of uniform amplitude excitation are the golden angle and the optimization of the angular separations. The cases of non-uniform amplitude excitations consider a raised cosine distribution and the optimization of the amplitude excitations through the Fermat spiral array. In this study, the method of genetic algorithms (GA) was used in the cases to find the values of the angular separations and the amplitude excitations of the Fermat spiral array. A performance evaluation was conducted for all these design cases, considering the Fermat spiral geometry. These design cases were validated using electromagnetic simulation to take mutual coupling into account and consider the effect of the antenna element pattern in each proposed design case.

## 1. Introduction

Antenna arrays are a key component for improving the performance of the communications systems in different applications for the new generation [1,2,3,4,5,6]. The design of antenna arrays includes aspects such as [7] the scanning range of the main beam, the amplitude excitation distribution, the number of elements and the geometry, among others. 

In antenna arrays, an open problem is to set the geometry with the best performance for a specific application. Many geometries of antenna arrays [8,9,10] have been proposed in the literature for different applications. In many applications, the large number of antenna elements needed to reach a desired performance is really a challenge. Furthermore, the apparition of grating lobes is another well-known challenge to be solved [11,12]. 

There have been many design techniques proposed in the state of art to solve these design challenges. The main techniques presented in the previous works have been based on aperiodic antenna arrays [11,13,14,15]. The aperiodic antenna arrays provide design options to solve the problem of grating lobes, with an increased performance. However, the design or synthesis of aperiodic antenna arrays could be a very complex problem, considering different structures or design variables.

Furthermore, the clustered array method was analyzed and studied in [16,17]. Specifically, in this method the array radiating elements lying on a regular and periodic lattice are divided into different clusters or sub-arrays, fed by a single transit/receive module (TRM). Some interesting examples about that array architecture were reported in [16,17] to solve the problem of grating lobes in the application of sub-arrays to reduce the number of active devices in the antenna system. 

The Fermat spiral [18] has demonstrated in the state of art that it provides an interesting performance, avoiding the apparition of grating lobes and obtaining a good SLL performance for widely spaced elements [19,20]. Although the geometry of the Fermat spiral has been studied previously in antenna arrays, the evaluation and new design techniques for this kind of structure are still scarce for SLL performance.

Therefore, this paper presents novel design techniques for the Fermat Spiral, considering a maximum SLL reduction. The array system in this study was based on a Fermat spiral configuration and the proposed design cases considered were based on uniform and non-uniform excitation. The design cases using a uniform amplitude excitation were the golden angle and the case of optimizing the angular separations. The design cases using a non-uniform amplitude excitation were the application of a raised cosine distribution and the optimization of the amplitude excitations through the Fermat spiral array. Furthermore, the method of genetic algorithms (GA) [13,21] was used in the proposed cases to find the values of the angular separations and the amplitude excitations of the Fermat spiral array. 

The novel contribution of this paper is the performance evaluation of all these proposed design cases, considering the Fermat spiral geometry. These design cases were validated using electromagnetic simulation to take mutual coupling into account and consider the effect of the antenna element pattern in each proposed design case.

## 2. Design Configurations for the Fermat Spiral Array

There are many possibilities for spiral geometries. However, the Fermat spiral presents an interesting property: a constant mean distance between neighboring elements [22,23,24]. Therefore, we selected this array geometry to analyze and evaluate some proposed design techniques. 

The antenna elements can be distributed in the geometry of the Fermat spiral (as shown in Figure 1) using the following coordinates [20]:(1)$$\rho_{n}=\frac{d}{d_{14}}\sqrt{n}$$
(2)$$\phi_{n}=n\pi \left(3-\sqrt{5}\right)$$
where *ρ_n_* is the radial distance of the *n*th element; $\phi_{n}$ is the angle of two adjacent elements (through the spiral); $\pi \left(3-\sqrt{5}\right)$ is the golden angle and $d_{14}=\sqrt{5-4\cos\phi_{3}}$. Equation (1) can be set for a minimal distance between antennas, *d*.

Then, the total electric field radiated by the spiral array can be calculated using the following expression for far field [20,25]: (3)$$\vec{E}\left(\vec{r}\right)=\vec{E_{0}}\left(\vec{r}\right)\;\;\overset{N}{\underset{n=1}{\sum }}a_{n}e^{\left[jk\left(x_{n}\sin\theta \cos\phi +y_{n}\sin\theta \sin\phi +\beta_{n}\right)\right]}$$
with $\vec{E_{0}}\left(\vec{r}\right)$ as the antenna element pattern; *k* = 2π/*λ*, 𝜃 and ϕ as the elevation and azimuth angles; *a_n_* as the amplitude excitation of the *n*th element and *N* as the number of elements distributed in the positions (*x*_n_, *y*_n_), which are calculated as in [20], as follows:(4)$$x_{n}=\rho_{n}\cos\left(\phi_{n}\right)$$
(5)$$y_{n}=\rho_{n}\sin\left(\phi_{n}\right)$$
(6)$$\beta_{n}=-x_{n}\sin\theta_{0}\cos\phi_{0}-y_{n}\sin\theta_{0}\sin\phi_{0}$$

The previous work (cited in Section 1) only considered the design case using uniform excitation, with the distribution of the antenna elements generated by the golden angle. In this study, the following three design cases were proposed for the Fermat spiral: (1) the application of a raised cosine distribution for determining the amplitude excitations, (2) the optimization of the amplitude excitations for the Fermat spiral array and (3) the optimization of the angular separation values to set the antenna elements through the Fermat spiral. The design methodology applied in each case is explained in the next sections. 

### 2.1. Raised Cosine Distribution

The application of a raised cosine distribution in antenna arrays follows the rule of Equation (7), as follows:(7)$$I_{n}=\frac{1+\cos\left(\frac{d_{n,m}\cos^{-1}\left(2a-1\right)}{0.5L}\right)}{2},\;\;n=1,2,3,\ldots N$$
where *d_n_* is the distance from the array center to the *n*th element, *L* is the array longitude over *x* axis and *a* is a constant value. The value of *I_n_* is assigned to be in the range of *a* < *I_n_* < 1.

In our design case, the values of the raised cosine distribution were set as indicated in Figure 2. A value of amplitude excitation was assigned to several elements within a specific area (or ring). Then, we controlled the amplitude values to try and minimize the SLL. The value of *L* was calculated as the distance from the array center to the farthest antenna element in the array. The index, *n*, in the value of *d_n_* indicates the area (ring) level for the *n*th element. Therefore, the value of *d_n_* was calculated by the difference or separation between the ring *n* and the ring *n* − 1.

### 2.2. Optimization of the Amplitude Excitations for the Fermat Spiral Array

The next design case was the optimization of the amplitude excitations for the Fermat spiral, using evolutionary optimization. The decision variables (amplitude excitations) were set in a vector of real numbers, *I* = [*I*_1_, *I*_2_, *I*_3_, *I*_4_, …, *I_N_*]. The design problem consisted of finding the set of amplitude excitation values that minimized the maximum SLL, considering a minimum distance, (*d_min_* ≥ *d_desired_*), among the elements. The design case can be described as the minimization of the next objective [20], as follows:(8)$$OF=SLL_{\max}\left(AF\left(\theta ,\phi \right)\right)+abs\left(d_{min}-d_{desired}\right)$$
where *SLL_max_* is the maximum side lobe level obtained in all cuts of *ϕ* = [0, 2π] for *θ* = [−π/2, π/2]. It was considered in our design case that the evaluation of the radiation pattern for each cut of *ϕ,* in steps of 10⁰, sweeping in *θ* = [−π/2, π/2].

The optimization of the amplitude excitations was made by GA. This is because GA has been shown to be effective in the design of antenna arrays [11,26,27]. The implementation of GA followed the methodology described in [11]. Please note, that this design case considered a different value of amplitude excitation for each antenna element, as shown in Figure 3.

### 2.3. Optimization of the Angular Separations through the Fermat Spiral Array

The last design case was the optimization of the angular separations through the Fermat spiral array, as shown in Figure 4. The decision variables (angular separations) were set in a vector of real numbers of angular separations, in radians α = [*α*_1_, *α*_2_, *α*_3_, *α*_4_, …, *α_N_*]. The values of the amplitude excitations (*I*_1_, *I*_2_, *I*_3_, *I*_4_, …, *I_N_*) were set to be equal to the unity. The design problem consisted of finding the set of angular separation values through the Fermat spiral array, which minimized the maximum SLL and also considered a minimum distance (*d_min_* ≥ *d_desired_*) among elements. This design case tried to minimize the objective function, *OF*, described in the Equation (8) of the previous section. The minimization of the *SLL_max_* was attained in all cuts of *ϕ* = [0, 2π] for *θ* = [−π/2, π/2]. As in the previous case, this design case considered the evaluation of the radiation pattern for each cut of *ϕ,* in steps of 10^0^, sweeping in *θ* = [−π/2, π/2]. Furthermore, the optimization of the angular separations through the Fermat spiral array was made by GA, considering the effectiveness of this method to design aperiodic antenna arrays [11]. A detailed description of the method of GA is provided in [11]. 

## 3. Results and Discussion

The three proposed design cases for Fermat spiral arrays were implemented to evaluate the maximum SLL reduction. The results of the next cases are illustrated here and were studied in order to make a comparative analysis among the different design cases for the Fermat spiral array, as follows: (a) the uniform case using the golden angle, (b) the optimization of the angular separations using uniform amplitude excitation, (c) the application of raised cosine for the amplitude distribution and (d) the optimization of the amplitude excitations. The uniform case, using the golden angle for the Fermat spiral array, has been analyzed previously in [20]. We followed the methodology described in the previous sections to implement the proposed cases of the raised cosine and the optimization of the amplitudes and the angular separations for the Fermat spiral array. The optimization parameters of GA were set following the literature and the previous experience of the authors of [26,27]. A crossover probability of 1.0 was used and the mutation probability was set as 0.1. The optimization algorithm was run in a number of iterations (1000), set to reach convergence. The population size was set in a value of twice the number of the decision variables considered in the design problem. All the design cases considered a value of *N* = 16 antenna elements and a minimal distance of *d_min_* = 0.5λ.

### 3.1. Cases Using Uniform Amplitude Excitation: Golden Angle and Angular Separations Optimized

The results of the design cases for the Fermat spiral array using the golden angle and optimizing the angular separations are presented in this section. These cases use uniform amplitude excitation through the Fermat spiral array.

Figure 5 illustrates the array factor results and the geometry generated for the golden angle case. The evaluation of the SLL performance considered all the cuts of *ϕ* = [0, 2π]. As shown in Figure 5b, this design case reached an SLL reduction of −13.27 dB, considering all the cuts in *ϕ*—i.e., this was the highest value of SLL found, and this SLL value was found at the cut of *ϕ* = 90°. However, the lowest value of SLL found was −16.67 dB, at the cut of *ϕ* = 40°. It is interesting to note that low values of SLL were found without requiring amplitude excitations. This is an important characteristic or benefit of the Fermat spiral arrays, which needs to be exploited in antenna systems’ applications.

Figure 6 shows the results generated by the optimization of the angular separations, optimized by GA. Figure 6b illustrates the array factor results for all the cuts of *ϕ*. The highest value of SLL was −14.2 dB, at the cut of *ϕ* = 20°, and the lowest value found was −18.10 dB. There was an SLL performance improvement with respect to the case of the golden angle. This performance was reached by maintaining similar values of beam-width, as shown in Figure 6. The array geometry generated and presented in Figure 6a illustrates an aperture size or array dimensions very similar with respect to the golden angle.

### 3.2. Cases Using Non-Uniform Amplitude Excitation: Raised Cosine and Amplitude Excitations Optimized

This section presents the results of the design cases of the non-uniform amplitude excitation, using the raised cosine distribution and optimizing the amplitude excitations through the Fermat spiral array. The golden angle was used to generate the positions of the antenna elements in these proposed design cases.

Figure 7 shows the results obtained for the case of applying a raised cosine distribution for amplitude excitations. The value of *a* was set to be 0.30 for the raised cosine distribution and the evaluation of the SLL performance considered all the cuts of *ϕ* = [0, 2π], as in the previous case. The application of the raised cosine provided the highest value of the SLL as −15.02 dB (*ϕ* = 100°), and the lowest value of the SLL as −22.06 dB, at the cut of *ϕ* = 40°. It is very interesting to note that these SLL values were provided by using only four amplitude excitations. This could be an interesting aspect for antenna designers, when tradeoffs or restrictions could be set in the number of amplitude excitations used in the system.

Figure 8 illustrates the results generated by the optimization of the amplitude excitations using GA. This design case considered a different value for each antenna element, i.e., 16 amplitude excitations. Figure 8 shows that the optimization of the amplitude excitations provided the highest value of SLL = −17.06 dB (*ϕ* = 90°) and the lowest value of SLL = −20.86 dB, at the cut of *ϕ* = 170°. This was the case with the best performance in the SLL reduction with respect to the other design cases of the Fermat spiral array. However, this performance was obtained by using one amplitude excitation by the antenna element. This could also be of interest for antenna designers—i.e., having the lowest case of SLL.

As seen in the previous results, there are interesting tradeoffs between the proposed design cases. These results were summarized to be illustrated in a numerical way, as in Table 1. This Table shows a comparative analysis of the proposed techniques (with respect to the previous work published in [20]) for the Fermat spiral array, for maximum SLL reduction. The case of amplitude optimization is the best case, considering the maximum SLL performance. However, this case considered the application of a non-uniform excitation with 16 different amplitude values. The raised cosine case offered a good tradeoff between the SLL performance and the use of four amplitude excitations. As a matter of fact, there were some cuts of *ϕ* that the raised cosine case outperformed—the case of amplitude optimization (*ϕ* = {10°, 20°, 30°, 40°, 50°, 60°}). The cases of using uniform amplitude excitation (such as the golden angle and optimizing angular separations) provided an interesting tradeoff between the SLL performance obtained, considering that amplitude excitations were not required. Except for the cuts of *ϕ =* {50°, 60°, 70°, 80°, 120°, 130°, 140°, 150°}, the case of optimizing the angular separations outperformed the case of the golden angle, obtaining lower values of SLL. Furthermore, the case of optimizing the angular separations outperformed all the other design cases in the cut of *ϕ =* 90°. Therefore, the best tradeoff can be chosen for the antenna designers, depending on the cut and the design requirement.

### 3.3. Analysis Considering Cophasal Excitation for Beam-Steering

Next, we analyzed the case of applying cophasal excitation to the Fermat spiral array. The cophasal excitation can be applied using the expression indicated previously in (Equation (6)). Table 2 shows a comparative analysis of the proposed techniques, considering beam-steering. This analysis considered the evaluation of the SLL for the scanning range of each proposed technique. The radiation pattern was scanned in the elevation plane for the direction of *θ*_0_. Although only positive values of *θ*_0_ are illustrated in Table 2, the same or very similar values of the SLL were obtained for negative values of *θ*_0_. 

As observed in Table 2, the case of amplitude optimization was the best case, considering a minimum SLL value of −20.09 dB and a maximum SLL value of −14.8 dB, for a scanning range of ±25°. However, this case considered the application of a non-uniform excitation (16 amplitude values). The raised cosine case offered a scanning range of ±23°, with a minimum SLL of −19.3 dB and −14.6 dB as a maximum value (with four amplitude excitations). The case of using the golden angle offered a scanning range of ±22°. The interesting aspect of the golden angle case was that the SLL value remained uniform, at −14.8 dB, for all the scanning ranges. The case of optimizing the angular separations was the worst case for providing beam-steering. This case provided a scanning range of ±5° for a minimum SLL of −16.08 dB and −15.12 dB as a maximum. Some examples of the radiation pattern obtained for the maximum scanning direction for each proposed technique are illustrated in Figure 9.

### 3.4. Full-Wave Electromagnetic Simulations

The proposed design cases for the Fermat spiral array were electromagnetically characterized by using CST Microwave Studio. This was made to analyze the performance of the proposed design cases, considering the effects of mutual coupling among the elements. A circular patch was considered as the antenna element (in the full-wave simulations), with a design frequency at 6 GHz, and the next characteristics were as follows: *r* = 13.02 mm, *h* = 1.6 mm and $p^{\prime }$ = 2.07 [7]. The material or substrate used was FR4. 

The results obtained by the electromagnetic simulation, taking mutual coupling into account, and the effect of the antenna element pattern are shown in Figure 10 and Figure 11. The design cases of using uniform amplitude excitation (the golden angle and optimizing the angular separations) are illustrated in Figure 10. The design cases of raised cosine and the optimization of the amplitudes (non-uniform amplitude excitation) are shown in Figure 11. It is very interesting to note the effect of including the antenna element pattern and of taking mutual coupling into account in the proposed design cases for the Fermat spiral array. The electromagnetic simulations provided an SLL_max_ = −15.11 dB and an SLL_min_ = −19.26 dB for the golden angle; an SLL_max_ = −16.07 dB and SLL_min_ = −20.76 dB for the angle optimization; an SLL_max_ = −17.79 dB and SLL_min_ = −24.72 dB for the raised cosine; and an SLL_max_ = −19.69 dB and SLL_min_ = −23.49 dB for the amplitude optimization. The radiation characteristics (considering the SLL performance) in the electromagnetic simulation were better with respect to the array factor results. The case of amplitude optimization remained the best case, considering the maximum SLL performance. However, there were some interesting aspects to note in the electromagnetic evaluation of the proposed design cases for the Fermat spiral array. Table 3 illustrates the numerical results obtained by the electromagnetic simulation for each cut of *ϕ* and the SLL performance of each proposed design case. As illustrated in Table 3, the raised cosine case outperformed all the other cases in the cuts of *ϕ* = {10°, 20°, 30°, 40°, 50°}. Note that the raised cosine used only four amplitude excitations. Another interesting aspect is that the angle optimization case outperformed all the other design cases in the cut of *ϕ =* 80° and was very close (less of 1 dB) to the best case in the cuts of *ϕ =* {30°, 70°, 80°, 90°}. This design case represented a good design tradeoff, considering the SLL performance in all cuts of *ϕ* and that amplitude excitations are not required. If more SLL performance is required, the raised cosine case could provide it at the expense of using four different amplitude excitations for the 16 antenna elements of the array. The amplitude optimization case obtained the best SLL performance, at the expense of using 16 different amplitude excitations for the Fermat spiral array. However, there is only a significant difference (more than 1 dB) with respect to the other cases in the cuts of *ϕ =* {80°, 100°, 110°, 120°, 130°, 150°, 160°}. The antenna designer will have interesting information, considering the different tradeoffs provided by each proposed design for the Fermat spiral array. 

Table 4 illustrates the gain values obtained for each proposed design of the Fermat spiral array. It can be observed that similar values of gain were obtained for all the proposed techniques. The cases of the amplitude optimization and the raised cosine (non-uniform excitation) required us to sacrifice a little more of the gain to obtain the SLL performance previously illustrated. 

Finally, it is important to mention that all the proposed design cases presented a good matching by providing a reflection coefficient with values lower than −10 dB at the design frequency. Figure 12 illustrates the behavior of the reflection coefficients versus the frequency for each proposed design case of the Fermat spiral array.

## 4. Conclusions

This paper illustrated novel design techniques for the Fermat spiral array. These proposed design techniques were evaluated considering the maximum SLL performance, by restraining the minimal separation among the antenna elements. The results of each proposed design case were analyzed considering uniform and non-uniform amplitude excitation. Each design case was validated by using electromagnetic simulations to take mutual coupling into account and considering the effect of the antenna element pattern. It was concluded that the case of amplitude optimization was the best case when considering the maximum SLL performance. However, this case considered the application of a non-uniform excitation, with 16 different amplitude values. The raised cosine case offered a good tradeoff between the SLL performance and the use of four amplitude excitations. The raised cosine case outperformed all the other cases in the cuts of *ϕ* = {10°, 20°, 30°, 40°, 50°}. Furthermore, the case of the optimization of the angular separations outperformed all the other design cases in the cut of *ϕ =* 80° and was very close (less of 1 dB) to the best case in the cuts of *ϕ =* {30°, 70°, 80°, 90°}. This design case represents a good design tradeoff, considering the SLL performance and that amplitude excitations are not required. Every design case presented a good matching, by providing a reflection coefficient with values lower than −10 dB at the design frequency. 

The antenna designer will have interesting information, considering the different tradeoffs provided by each proposed design for the Fermat spiral array. 

## Figures and Tables

**Figure 1 micromachines-13-02000-f001:**
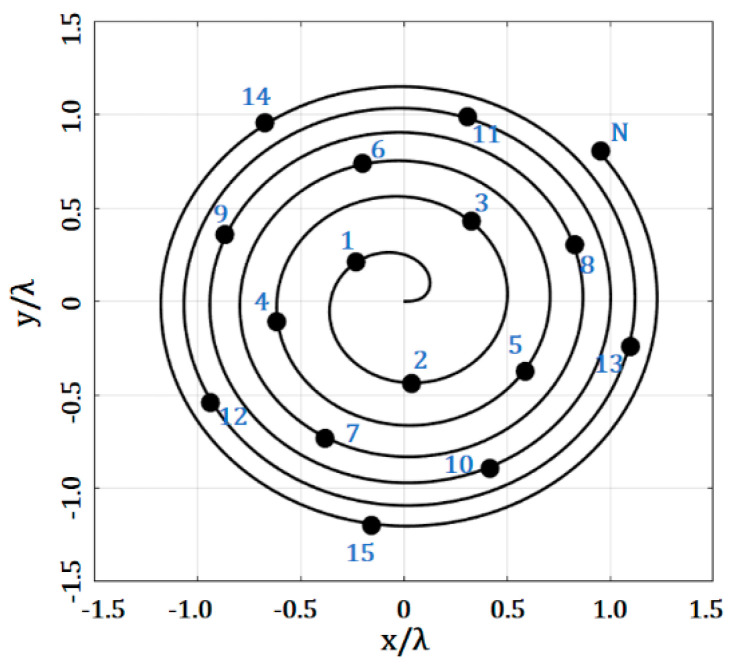
Antenna array, with elements distributed in the geometry of the Fermat spiral.

**Figure 2 micromachines-13-02000-f002:**
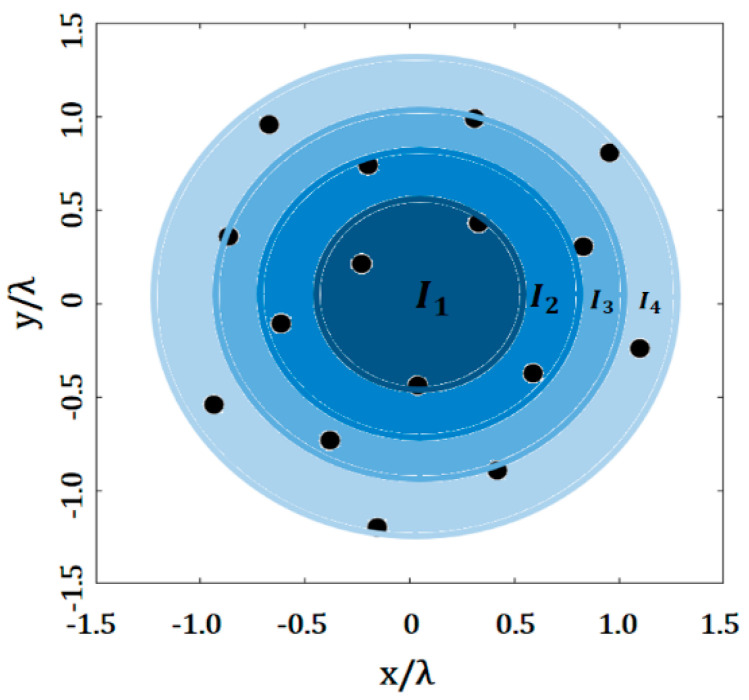
Application of the raised cosine distribution to the Fermat spiral.

**Figure 3 micromachines-13-02000-f003:**
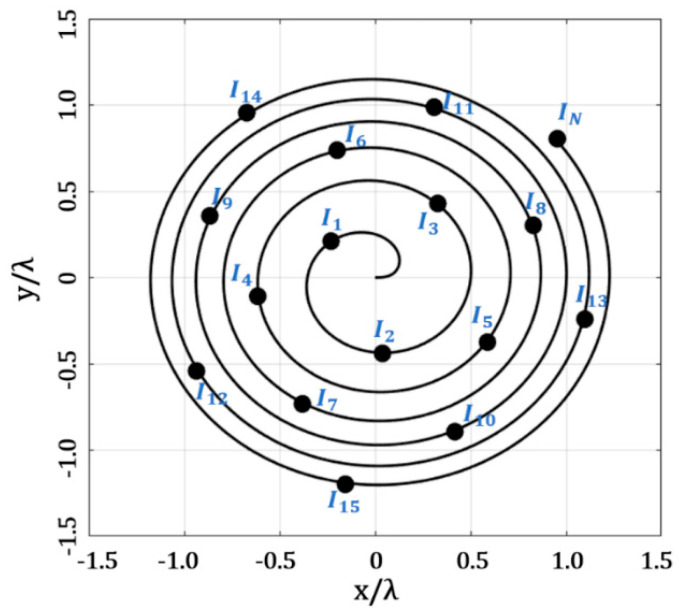
Application of GA to optimize the amplitude excitations of the Fermat spiral array.

**Figure 4 micromachines-13-02000-f004:**
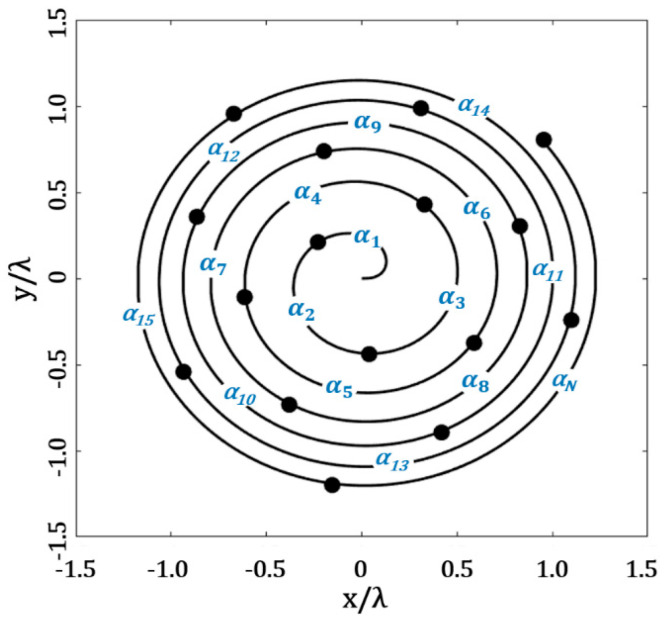
Application of GA to optimize the angular separations of the Fermat spiral array.

**Figure 5 micromachines-13-02000-f005:**
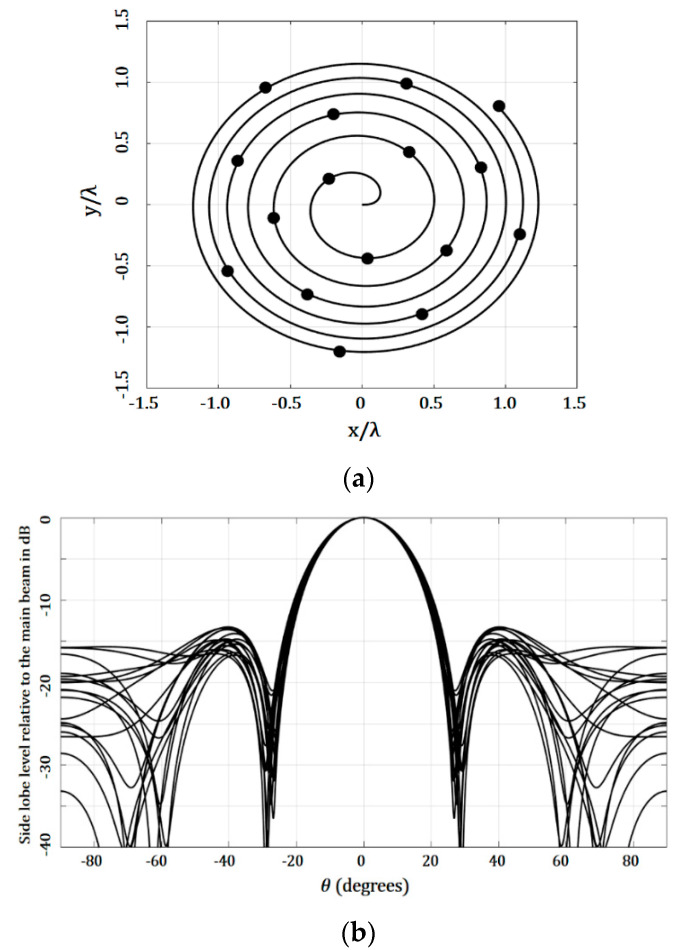
Results for the design case using the golden angle (uniform amplitude excitation): (**a**) geometry generated and (**b**) array factor.

**Figure 6 micromachines-13-02000-f006:**
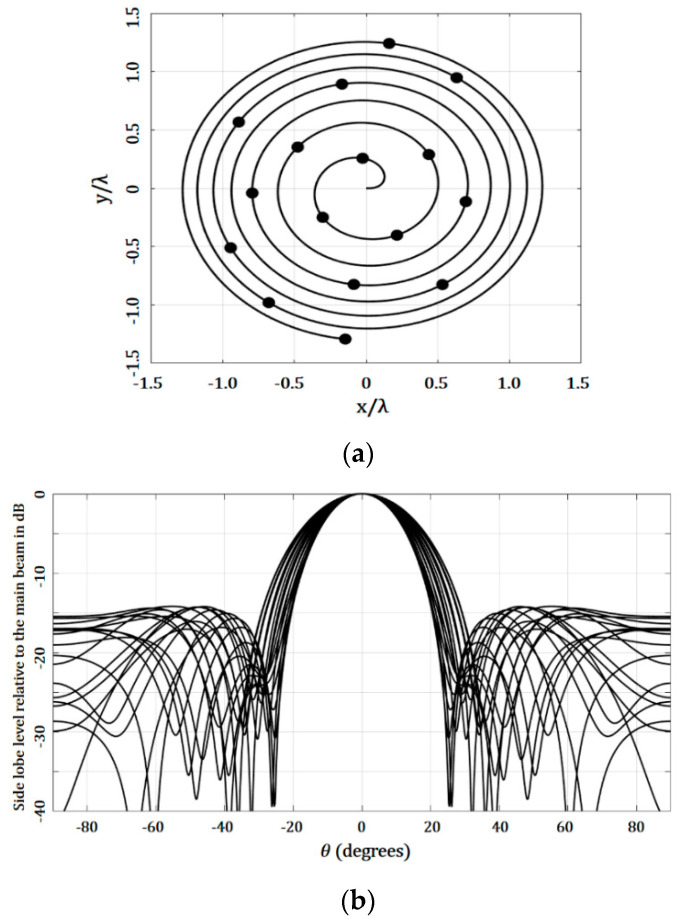
Results for the design case optimizing the angular separations using GA (uniform amplitude excitation): (**a**) geometry generated and (**b**) array factor.

**Figure 7 micromachines-13-02000-f007:**
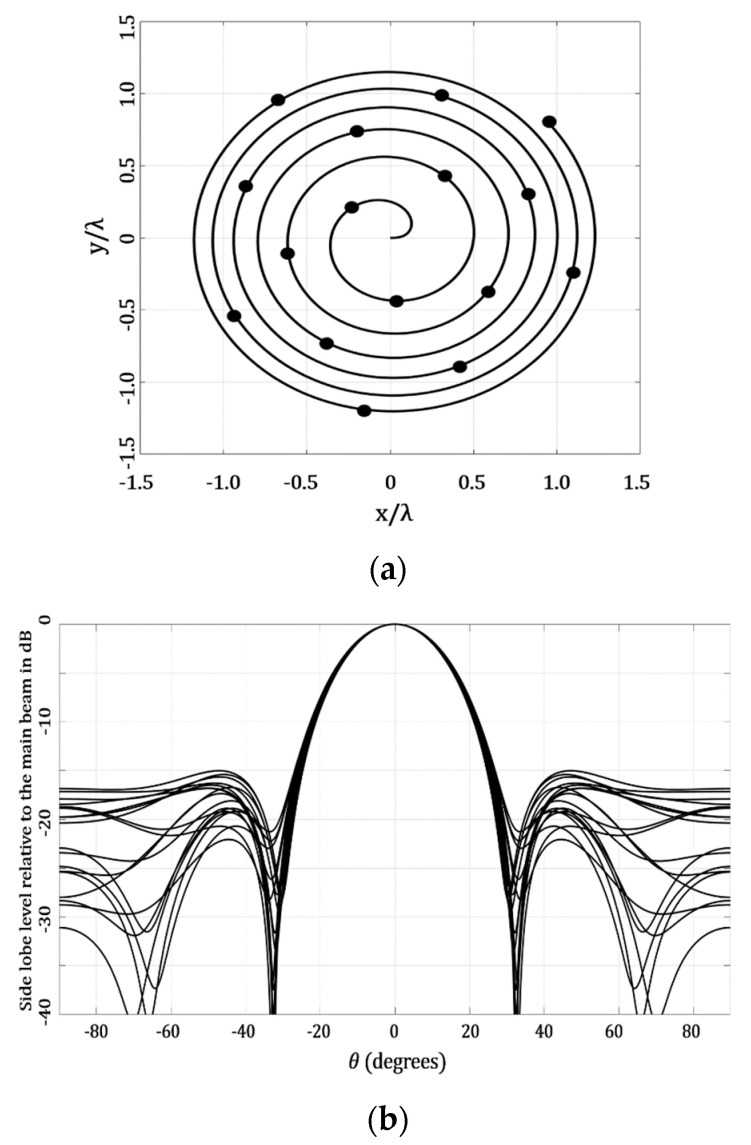
Design case that considers the application of raised cosine for a non-uniform amplitude excitation: (**a**) geometry generated and (**b**) array factor results.

**Figure 8 micromachines-13-02000-f008:**
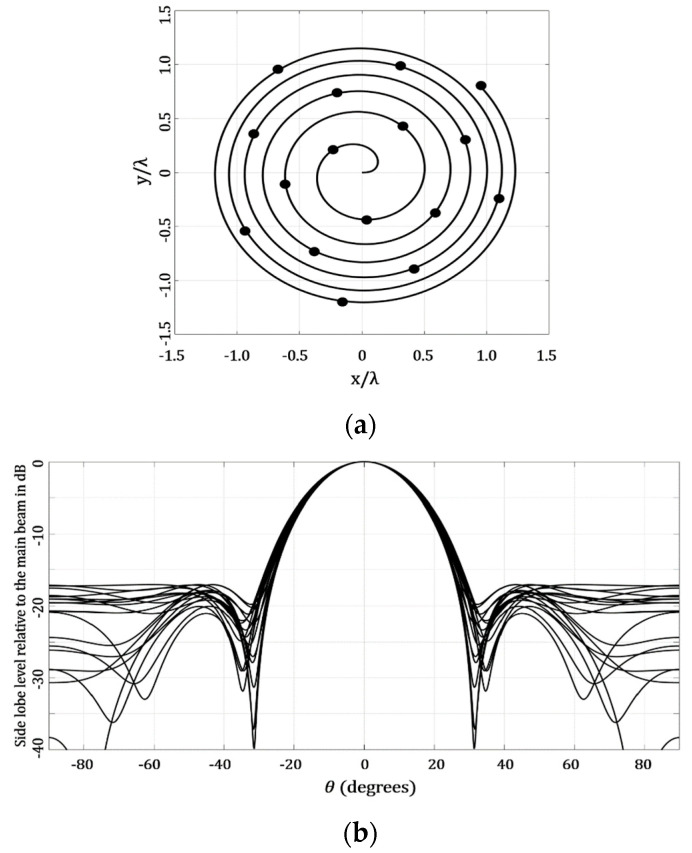
Design case that considers the optimization of the amplitude excitations using GA: (**a**) geometry generated and (**b**) array factor results.

**Figure 9 micromachines-13-02000-f009:**
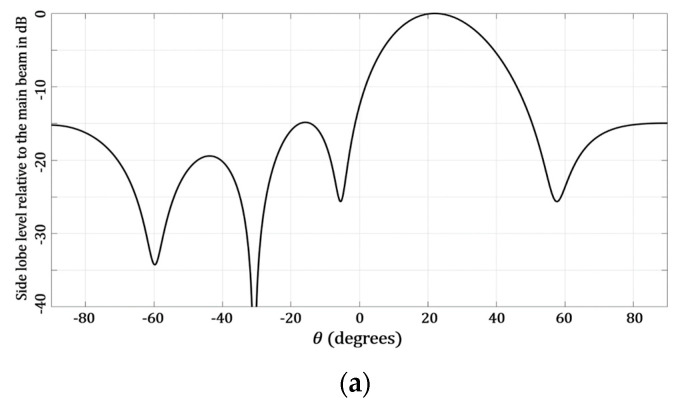
Radiation pattern obtained for the maximum scanning direction for the following: (**a**) golden angle, (**b**) angle optimization, (**c**) raised cosine and (**d**) amplitude optimization.

**Figure 10 micromachines-13-02000-f010:**
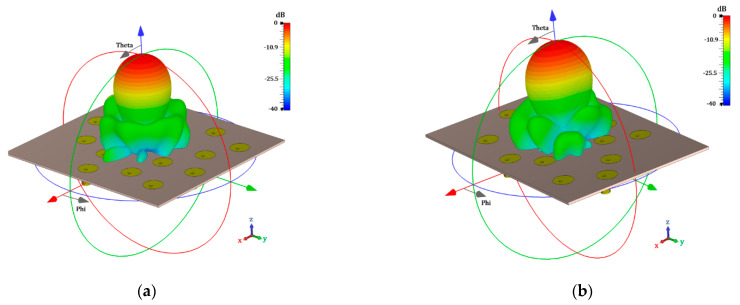
Radiation pattern obtained by full-wave simulation for the cases of using uniform amplitude excitation, as follows: (**a**) golden angle and (**b**) optimization of angular separations.

**Figure 11 micromachines-13-02000-f011:**
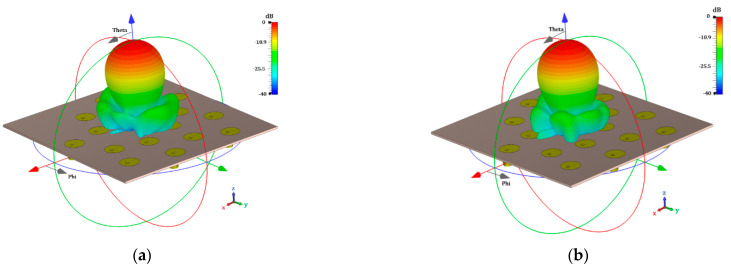
Radiation pattern obtained by full-wave simulation for the cases of using non-uniform amplitude excitation, as follows: (**a**) raised cosine and (**b**) optimization of amplitude excitations.

**Figure 12 micromachines-13-02000-f012:**
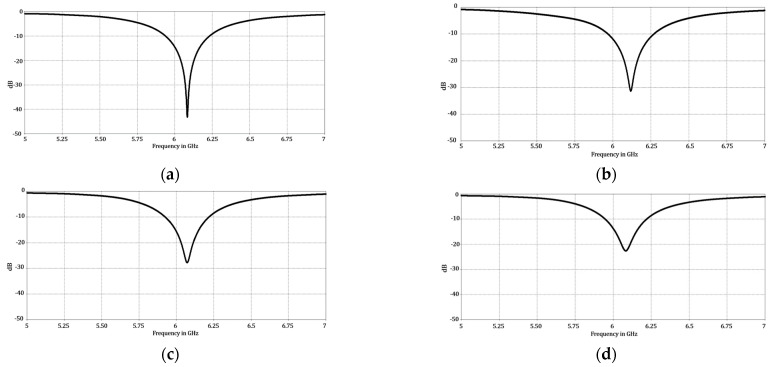
Behavior of the reflection coefficient versus frequency for all proposed design cases of the Fermat spiral, as follows: (**a**) golden angle, (**b**) angle optimization, (**c**) raised cosine and (**d**) amplitude optimization.

**Table 1 micromachines-13-02000-t001:** A comparative analysis of the proposed techniques, with respect to the previous work published in [20], for the Fermat spiral array for maximum SLL reduction (array factor results).

Design Case	Amplitude Excitation	SLL in dB for Each Cut of 𝝓								
		10°	20°	30°	40°	50°	60°	70°	80°	90°
Golden angle [20]	Uniform	−13.57	−13.52	−15.00	−16.67	−16.37	−15.34	−14.75	−14.06	−13.27
Angle optimization	Uniform	−14.28	−14.22	−16.83	−17.22	−15.12	−14.61	−14.41	−15.41	−18.10
Raised cosine	Non-uniform	−18.88	−19.18	−20.71	−22.06	−18.85	−18.70	−18.10	−16.73	−15.41
Amplitude optimization	Non-uniform	−18.18	−17.07	−17.09	−17.87	−17.55	−18.58	−18.95	−18.01	−17.06
**Design Case**	**Amplitude Excitation**	**SLL in dB for Each Cut of** **𝝓**								
		**100°**	**110°**	**120°**	**130°**	**140°**	**150°**	**160°**	**170°**	**180°**
Golden angle [20]	Uniform	−13.41	−14.74	−15.79	−15.67	−16.01	−14.84	−14.86	−15.52	−14.84
Angle optimization	Uniform	−16.82	−15.69	−15.07	−14.62	−14.30	−14.18	−15.44	−17.06	−16.08
Raised cosine	Non-uniform	−15.02	−15.66	−16.30	16.81	−16.83	−16.50	−17.36	−18.87	−19.30
Amplitude optimization	Non-uniform	−17.13	−18.23	−18.23	−17.51	−17.07	−18.24	−20.11	−20.86	−20.09

**Table 2 micromachines-13-02000-t002:** A comparative analysis of the proposed techniques, considering beam-steering.

Array Type		SLL (dB) for Each Value of *θ*_0_														
		1°	2°	3°	4°	5°	6°	7°	8°	9°	10°	11°	12°	13°	14°	15°
Golden angle		−14.84	−14.84	−14.84	−14.84	−14.84	−14.84	−14.84	−14.84	−14.84	−14.84	−14.84	−14.84	−14.84	−14.84	−14.84
Angle optimization		−16.08	−16.08	−16.08	−16.08	−15.12	−14.08	−13.18	−12.39	−11.71	−11.11	−10.59	−10.13	−9.74	−9.4	−9.13
Raised cosine		−19.30	−19.30	−19.30	−19.30	−19.30	−19.30	−19.30	−19.30	−19.30	−19.30	−19.30	−19.30	−19.30	−19.30	−19.30
Amp. optimization		−20.09	−20.09	−20.09	−20.09	−20.09	−20.09	−20.09	−20.09	−20.09	−20.09	−20.09	−20.09	−20.09	−20.09	−20.09
**Array Type**		**SLL (dB) for Each Value of *θ*_0_**														
		**16°**	**17°**	**18°**	**19°**	**20°**	**21°**	**22°**	**23°**	**24°**	**25°**	**26°**	**27°**	**28°**	**29°**	**30°**
Golden angle		−14.84	−14.84	−14.84	−14.84	−14.84	−14.84	−14.84	−14.26	−13.43	−12.7	−12.06	−11.50	−11.02	−10.60	−10.24
Angle optimization		−8.91	−8.73	−8.60	−8.51	−8.46	−8.45	−8.48	−8.45	−8.45	−8.45	−8.45	−8.45	−8.45	−8.45	−8.45
Raised cosine		−19.30	−19.30	−18.78	−17.8	−16.89	−16.06	−15.31	−14.63	−14.02	−13.46	−12.96	−12.52	−12.12	−11.77	−11.45
Amp. optimization		−20.09	−20.09	−20.09	−20.09	−18.98	−17.97	−17.05	−16.22	−15.47	−14.80	−14.19	−13.65	−13.17	−12.74	−12.36

**Table 3 micromachines-13-02000-t003:** Numerical results obtained by the electromagnetic simulation for each cut of *ϕ* and the SLL performance of each proposed design case of the Fermat spiral array.

Design Case	Amplitude Excitation	SLL in dB for Each Cut of 𝝓								
		10°	20°	30°	40°	50°	60°	70°	80°	90°
Golden angle	Uniform	−15.93	−15.73	−17.51	−18.56	−18.47	−17.33	−16.56	−15.67	−15.11
Angle optimization	Uniform	−16.51	−16.07	−18.89	−20.07	−19.38	−19.98	−20.01	−20.76	−18.65
Raised cosine	Non-uniform	−22.54	−23.3	−23.86	−24.72	−24.13	−21.71	−20.33	−19.03	−18.01
Amplitude optimization	Non-uniform	−21.50	−20.75	−19.98	−19.92	−21.04	−22.02	−21.21	−20.29	−19.69
**Design Case**	**Amplitude Excitation**	**SLL in dB for Each Cut of** **𝝓**								
		**100°**	**110°**	**120°**	**130°**	**140°**	**150°**	**160°**	**170°**	**180°**
Golden angle	Uniform	−15.61	−17.18	−17.77	−19.26	−17.90	−16.46	−16.42	−17.49	−17.22
Angle optimization	Uniform	−17.69	−17.70	−17.92	−17.85	−17.43	−17.19	−18.97	−20.53	−19.18
Raised cosine	Non-uniform	−17.79	−18.38	−19.33	−20.12	−19.79	−19.52	−20.35	−22.27	−22.50
Amplitude optimization	Non-uniform	−19.86	−20.93	−21.65	−21.55	−21.05	−22.09	−22.79	−23.49	−22.96

**Table 4 micromachines-13-02000-t004:** Gain values obtained by the electromagnetic simulation for each proposed design case of the Fermat spiral array.

Proposed Technique	Gain (dB)
Golden angle	16.57
Angle optimization	16.54
Raised cosine	16.07
Amplitude optimization	15.93

## Data Availability

Not applicable.

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
