# Peer review of "Novel Design Techniques for the Fermat Spiral in Antenna Arrays, for Maximum SLL Reduction"

_micromachines, 2022, doi:10.3390/mi13112000_

Round 1

Reviewer 1 Report

The submitted paper presents a design technique for the Fermat Spiral considering a maximum SLL reduction. The topic and the proposed idea are interesting. My comments are reported hereafter:

In the paper introduction the authors mentioned just the aperiodic antenna arrays technique to solve some design challenges. For the sake of readers, it would be also interesting to mention the clustered array method. Specifically, in this method the array radiating elements lying on a regular and periodic lattice are divided in different clusters or subarrays fed by a single transit/receive module (TRM). Some very interesting examples about this unconventional array architecture that could be introduced in the proposed paper introduction are reported in [1R]-[2R].

[1R] F. A. Dicandia and S. Genovesi, ‘Wide-Scan and Energy-Saving Phased Arrays by Exploiting Penrose Tiling Subarrays’, IEEE Transactions on Antennas and Propagation, pp. 1–1, 2022, doi: 10.1109/TAP.2022.3178917.

[2R] P. Rocca, R. J. Mailloux, and G. Toso, ‘GA-Based Optimization of Irregular Subarray Layouts for Wideband Phased Arrays Design’, IEEE Antennas and Wireless Propagation Letters, vol. 14, pp. 131–134, 2015, doi: 10.1109/LAWP.2014.2356855.

When the authors introduce the Fermat spiral array, I suggest also to mention the following interesting paper to make a more precise literature overview [3R].

A. Ramalli, E. Boni, A. S. Savoia, and P. Tortoli, ‘Density-tapered spiral arrays for ultrasound 3-D imaging’, IEEE Transactions on Ultrasonics, Ferroelectrics, and Frequency Control, vol. 62, no. 8, pp. 1580–1588, Aug. 2015, doi: 10.1109/TUFFC.2015.007035.

In section 3 (Results and discussion) the comparison has been performed just highlighting the SLL. It should be interesting to introduce the array gain value. Please introduce in section 3 the array gain value.

 Another lack of the performed results is the absence of the beam steering. Please make a comparative analysis among the different design cases also in case a main beam steering.

Author Response

Response to Reviewer 1 Comments

The submitted paper presents a design technique for the Fermat Spiral considering a maximum SLL reduction. The topic and the proposed idea are interesting. My comments are reported hereafter:

  1. In the paper introduction the authors mentioned just the aperiodic antenna arrays technique to solve some design challenges. For the sake of readers, it would be also interesting to mention the clustered array method. Specifically, in this method the array radiating elements lying on a regular and periodic lattice are divided in different clusters or subarrays fed by a single transit/receive module (TRM). Some very interesting examples about this unconventional array architecture that could be introduced in the proposed paper introduction are reported in [1R]-[2R].

[1R] F. A. Dicandia and S. Genovesi, ‘Wide-Scan and Energy-Saving Phased Arrays by Exploiting Penrose Tiling Subarrays’, IEEE Transactions on Antennas and Propagation, pp. 1–1, 2022, doi: 10.1109/TAP.2022.3178917.

[2R] P. Rocca, R. J. Mailloux, and G. Toso, ‘GA-Based Optimization of Irregular Subarray Layouts for Wideband Phased Arrays Design’, IEEE Antennas and Wireless Propagation Letters, vol. 14, pp. 131–134, 2015, doi: 10.1109/LAWP.2014.2356855.

Response: The paper was corrected to include the clustered array method in the Introduction Section. A description of this array architecture was added, please see the revised version (pages 1-2). Furthermore, the advised references were added to the revised version of our manuscript.

  1. When the authors introduce the Fermat spiral array, I suggest also to mention the following interesting paper to make a more precise literature overview [3R].
  2. Ramalli, E. Boni, A. S. Savoia, and P. Tortoli, ‘Density-tapered spiral arrays for ultrasound 3-D imaging’, IEEE Transactions on Ultrasonics, Ferroelectrics, and Frequency Control, vol. 62, no. 8, pp. 1580–1588, Aug. 2015, doi: 10.1109/TUFFC.2015.007035.

Response: The paper was corrected to include the interesting recommended paper. That paper was cited when the Fermat spiral array was introduced to make a more precise literature overview. Please see the revised version of our paper (page 2 Introduction Section).

  1. In section 3 (Results and discussion) the comparison has been performed just highlighting the SLL. It should be interesting to introduce the array gain value. Please introduce in section 3 the array gain value.

Response: The paper was corrected to include the array gain value. Please see Table 4 page 12 of the revised version of our paper.

  1. Another lack of the performed results is the absence of the beam steering. Please make a comparative analysis among the different design cases also in case a main beam steering.

Response: The paper was corrected to include a comparative analysis considering beam-steering for the proposed techniques. Please see pages 9-10 of the revised version of our paper.

Thanks for your comments!

Reviewer 2 Report

The authors presented an interesting work on Novel Design Techniques for the Fermat Spiral in Antenna Arrays for Maximum SLL Reduction. The originality and novelty of the paper is high. The paper soundness is quite scientific which will results in more interest of the reader. However, the presentation lacks in many aspects which need to be updated before making final decision. Therefore, authors are requested to revise manuscript for the grammatical mistakes and typos. Authors should cite the source of all the equations properly. It is recommended to replace the  outdated reference with the state-of-the-art including on-demand frequency switchable antenna array operating at 24.8 and 28 ghz for 5G high-gain sensors applications, Helix inspired 28 GHz broadband antenna with end-fire radiation pattern, Simple wideband extended aperture antenna-inspired circular patch for V-band communication systems. The size of all picture should be kept same and small, some pictures are too large and result in lowering the presentation of the manuscript. Why there are so many graphs in Fig. 5(b), 6(b), 7(b) and 8(b), if they are showing different value to add legends and change the color of each graph and line shape to distinguish them. Reduce the size of Fig. 9 and 10 to make it look more scientific and reduce the unnecessary page length. Please redraw fig 11 to improve its readability. Paper should be formatted according to journal provided template. All the references should be re-written in journal template. 

Author Response

Response to Reviewer 2 Comments

The authors presented an interesting work on Novel Design Techniques for the Fermat Spiral in Antenna Arrays for Maximum SLL Reduction. The originality and novelty of the paper is high. The paper soundness is quite scientific which will results in more interest of the reader. However, the presentation lacks in many aspects which need to be updated before making final decision.

  1. Therefore, authors are requested to revise manuscript for the grammatical mistakes and typos.

Response: The paper was corrected and revised for the grammatical mistakes and typos. Please see the revised version of our manuscript.

  1. Authors should cite the source of all the equations properly.

Response: The paper was corrected and revised to cite the source of all the equations properly. Please see the revised version of our manuscript.

  1. It is recommended to replace the outdated reference with the state-of-the-art including on-demand frequency switchable antenna array operating at 24.8 and 28 ghz for 5G high-gain sensors applications, Helix inspired 28 GHz broadband antenna with end-fire radiation pattern, Simple wideband extended aperture antenna-inspired circular patch for V-band communication systems.

Response: The paper was corrected and revised to cite the source of all the equations properly. Please see the revised version of our manuscript.

  1. The size of all picture should be kept same and small, some pictures are too large and result in lowering the presentation of the manuscript.

Response: The paper was corrected to keep a standard format for the size of all pictures.

  1. Why there are so many graphs in Fig. 5(b), 6(b), 7(b) and 8(b), if they are showing different value to add legends and change the color of each graph and line shape to distinguish them.

Response: That is because the evaluation of the SLL performance considers all the cuts of f=[0,2p]. We only consider only one color, as the objective is to illustrate the highest value of SLL for all the cuts. Furthermore, as there are many cuts of f, it could be a little impractical to add many legends in the figures.

  1. Reduce the size of Fig. 9 and 10 to make it look more scientific and reduce the unnecessary page length.

Response: The paper was corrected to reduce the size of those figures. Please see the revised version of our manuscript.

  1. Please redraw fig 11 to improve its readability.

Response: That figure was redraw. Please see the revised version of our manuscript.

  1. Paper should be formatted according to journal provided template. All the references should be re-written in journal template.

Response: The paper was corrected to be formatted according to journal template. The references are in the journal format.

 Thanks for your comments!

Round 2

Reviewer 2 Report

The authors have addressed all the comments properly. Thus, the manuscript is recommended for publication.